# Effect of Salt Stress on Microbiome Structure and Diversity in Chamomile (*Matricaria chamomilla* L.) Rhizosphere Soil

Fei Xia [1,2,†], Haiping Hao [3,†], Ying Qi [3], Hongtong Bai [1,2], Hui Li [1,2,*], Zhenxia Shi [3,*] and Lei Shi [1,2,*]

[1] Key Laboratory of Plant Resources, Institute of Botany, Chinese Academy of Sciences, No. 20 Nanxincun, Xiangshan, Beijing 100093, China; 13651377226@163.com (F.X.)
[2] China National Botanical Garden, No. 20 Nanxincun, Xiangshan, Beijing 100093, China
[3] College of Life Science, Langfang Normal University, No. 100 Aiminxi Dao, Langfang 065000, China; haohaiping2014@126.com (H.H.); qiying_doc@163.com (Y.Q.)
[*] Correspondence: lihui@ibcas.ac.cn (H.L.); shizhenxia@lfnu.edu.cn (Z.S.); shilei@ibcas.ac.cn (L.S.); Tel.: +86-10-62836054 (H.L.); +86-0316-2188390 (Z.S.); +86-10-62836270 (L.S.)
[†] These authors contributed equally to this work.

**Abstract:** Chamomile (*Matricaria chamomilla* L.) is an economically valuable plant with certain salt alkali adaptability. Here, we aim to understand how salt stress affects both the structure and diversity of the soil microbial community and how root exudates may mediate this response. The results showed that high salt stress treatment reduced the overall diversity and abundance of both bacteria and fungi but did not alter the presence or abundance of dominant phyla, including Proteobacteria, Acidobacteriota, and Ascomycota. Several microbial species belonging to *Geminicoccaceae*, *Rokubacteriaces*, and *Funneliformis*-sp were found to be highly resistant to salt stress, while others were found to be highly sensitive, including *Xanthobacteraceae*, JG30-KF-AS9-sp, and *Asperellum*. Redundancy analysis results showed that bacteria tended to be more sensitive to the presence of salt ions in the soil, including $SO_4^{2-}$, $Ca^{2+}$, and $Na^+$, while fungi were more sensitive to the presence of certain root exudates, including methyl 4-methylbenzoate, δ-selinene. It suggested that the presence of a relatively stable set of dominant phyla and the increased abundance of salt-tolerant species and their ecological functions may be related to the tolerance of chamomile to salt stress. The results will underpin future improvement in chamomile to coastal salinity soil tolerance through altering the soil microbial community.

**Keywords:** chamomile; rhizosphere microbial; root exudation; salt stress





## 1. Introduction

Soil and aquifer salinization is fast becoming one of the most serious threats to agriculture worldwide. Currently, approximately 7% of the earth's land surface (including 20% cultivable fields and 70% dry land) is composed of saline soil, and the amount of affected land is increasing [1]. Excess salt ions $Na^+$, $SO_4^{2-}$, $Ca^{2+}$, $Mg^{2+}$, and alkaline pH threaten the normal growth and development of plants. Plants adapt to salt stress primarily at the root level, and microbes present in the rhizosphere can significantly impact the ability of plant roots to adapt to salt stress [2]. Rhizosphere microbes can affect plant growth and productivity both directly and indirectly, including 1. regulating root nutrient uptake, various bacterial genera, such as *Bacillus*, *Azotobacter*, *Pantoea*, etc., can ensure zinc, phosphate, and potassium solubilization to promote plant absorption [3]; 2. promoting endogenous hormone homeostasis, hormones, such as auxin indole-3-acetic acid, cytokinins, and jasmonic acid produced by *Pseudomonas*, *Bacillus*, *Acinetobacter*, etc., can increase root proliferation, plant cell division, and the production of plant primary and secondary metabolites, thereby improving plant salt stress tolerance; and 3. regulating root water conductivity, antioxidant enzyme defense system activity, and other regulatory functions [4,5].

In agricultural systems, several promising rhizosphere microbes have been identified and utilized as inoculants to promote tolerance of salt stress in crop plants. In rice, inoculation with *Azospirillum brasilense* and *Pseudomonas fluorescens* promotes the mineralization of soil nitrogen, improves the nitrogen supply of the soil, and ultimately results in increased biomass production [6]. In wheat, inoculation with *Aeromonas hydrophila/A. caviae* MAS765, *Bacillus insolitus* MAS17, *Bacillus* sp. MAS620, and *Bacillus* sp. MAS820 promotes plant growth under salt stress by limiting sodium uptake [7]. In pepper, inoculation with *Arthrobacter* sp. and *Bacillus* sp. enhances salt stress tolerance by promoting the accumulation of the osmoregulatory amino acid proline in plant cells [8].

Saline–alkali toxicity is largely responsible for the physiological effects of salt stress on microorganisms [9]. Specifically, salinity increases the osmotic potential of the soil environment, resulting in toxicity when the osmoregulatory capacity of microorganisms is overwhelmed [10,11]. Studies have found that in some soils, bacterial abundance and diversity are negatively affected by increasing $Na^+$ and $K^+$ levels [10,12]. However, certain microbial clades and species have shown a stronger salt tolerance. For example, work by Zhang et al. showed that Bacteroidetes and Gemmatimonadetes are able to tolerate a wide range of soil salinities [13]. Additionally, the relative abundance of the fungal phylum Ascomycota has been found to increase in saline soils [14]. Studying the responses of these and other salt-responsive microbes to increase soil salinity will be crucial to understand microbe-mediated salt tolerance in plants.

Plants interact with soil microbes primarily through the production of root exudates, and the quantity and quality of these root exudates vary across environments [15]. NaCl stress significantly affected the composition of *Phragmites australis* root exudates, especially the amino acid content [16]. Organic molecules of root exudates, such as thehalose, proline, acetylated glutamine dipeptides, and carboxylamines, could be used as osmotics by halophilic microorganisms to protect cells from salt stress [17]. Under salt stress conditions, root exudates could promote *Ensifer meliloti CL 09* growth, which suggests that root exudates might be one of the factors for rhizosphere and endosphere microbial selection under salt stress [18]. Although the relationship between root exudates and root microbes is receiving increased attention, the effect of salt stress on the composition of root exudates and the subsequent effects on the soil microbial community are still unknown.

Chamomile (*Matricaria chamomilla* L.) is one of the oldest traditional herbal medicines and is widely cultivated for its flowers, essential oils, and extracts [19,20]. Chamomile oil is used in medicine, cosmetics, aromatherapy, and beverages. Its flowers are primarily used to make tea [21,22]. Chamomile is an economically valuable plant with a certain salt and alkali tolerance and is one of the alternative plant materials for the development and utilization of saline–alkali soil. In this study, we utilized the saline–alkali soil of the Yellow River Delta to study the response of the chamomile rhizosphere microbial community to salt stress. We aimed to 1. understand how salt stress affects both the structure and diversity of the soil microbial community; 2. explore how the rizosphere environment of chamomile mediates this response; and 3. promote the cultivation of chamomile in coastal salinity soil by altering the soil microbial community. The results of this study will provide a valuable resource for the continued study of microbe-mediated salt tolerance in plants.

## 2. Materials and Methods

### 2.1. Soil and Plant Materials and Treatments

Experimental soil (0–20 cm deep) was collected from a Yellow River Delta (YRD) wetland area with an annual mean air temperature of 12.4 °C, evaporation of 1928.2 mm, and precipitation of 511.6 mm. Soil was collected along a salinity gradient determined according to the distribution of vegetation species and electroconductivity ($EC_{se}$). The low-salinity wetland (LS, $EC_{se}$ = 1.84 $dSm^{-1}$) was dominated by *Calamagrostis pseudophragmites*, medium-salinity wetland (MS, $EC_{se}$ = 4.58 $dSm^{-1}$) was dominated by *Phragmites australis*, and high-salinity wetland (HS, $EC_{se}$ = 8.96 $dSm^{-1}$) was dominated by *Tamarix chinensis* and *Suaeda salsa*. The soil was transported to the greenhouse, a small portion was used for

determining chemical and physical properties (Table 1), and the others were sun-dried, then sieved and cleared of rocks, roots, and other debris as experiment medium. Experimental pots were filled with 3.5–4 kg of soil and kept under controlled conditions: temperature of $25 \pm 2$ °C, photosynthetically active radiation of $800 \pm 50$ µmol m$^{-2}$s$^{-1}$, and relative humidity of $75 \pm 5$ %.

**Table 1.** The chemical and physical properties of the soil sample.

| | pH | Soil Organic (mg·kg$^{-1}$) | Total N (mg·kg$^{-1}$) | Total P (mg·kg$^{-1}$) | Total K (mg·kg$^{-1}$) |
|---|---|---|---|---|---|
| HS | $8.96 \pm 0.04$ | $5.67 \pm 0.71$ | $0.23 \pm 0.07$ | $0.71 \pm 0.02$ | $2.5 \pm 0.02$ |
| MS | $8.21 \pm 0.27$ | $6.37 \pm 0.84$ | $0.23 \pm 0.05$ | $0.69 \pm 0.03$ | $2.1 \pm 0.32$ |
| LS | $8.38 \pm 0.18$ | $6.67 \pm 0.67$ | $0.26 \pm 0.06$ | $0.65 \pm 0.03$ | $1.4 \pm 0.12$ |

Chamomile seedlings (seeds were provided by the Dutch company Hem Zaden B.V.) were grown in standard potting soil until they had 3–5 true leaves, at which point the strongest and healthiest seedlings were transplanted into LS, MS, or HS soils. Each salinity level consisted of 10 pots, repeated 3 times. In order to mimic naturally occurring saline–water conditions, the bottom of each pot was lined with a tray, and each time the pots were watered (every 5 to 7 days, 200–300 mL each time) with deionized water, the excess water drained from each pot was re-applied to the pot where it originated. When chamomile plants were in full bloom, the experimental soil samples were collected for examination of root exudates and soil microbial communities.

To collect rhizosphere soil samples, plants were removed and shaken vigorously to dislodge the soil clinging to the roots. To collect soil directly from the root surface, 3 chamomile root systems were harvested from each soil treatment and placed individually in 50 mL centrifuge tubes containing 25 mL of phosphate buffer (PB, per liter: 6.33 g of NaH$_2$PO$_4$·H$_2$O, 16.5 g of Na$_2$HPO$_4$·H$_2$O, 2 mL of Silwet L-77). Samples were centrifuged for 15 min at 5000 r/min and then filtered using a 1 mm nylon mesh cell. The filtered solution was centrifuged again for 15 min at 5000 r/min, and the supernatant was subsequently frozen in liquid nitrogen and stored at $-80$ °C. Rhizosphere soil from the roots (10 g) was collected in sterilized centrifuge tubes and stored at $-80$ °C.

*2.2. DNA Sequencing and Microbial Community Characterization*

Genomic DNA was extracted from each soil sample using a soil DNA Isolation Kit (Shanghai Majorbio Bio-Pharm Technology Corporation, Shanghai, China) according to the manufacturer's instructions. Agarose gel electrophoresis was utilized to verify the integrity of the DNA. The bacterial 16S rRNA gene was amplified using the 515f and 806r primer set (5′-GTGCCAGCMGCCGCGGTAA-3′/5′-GGACTACHVGGGTWTCTAAT-3′) [21]. The fungal ribosomal internal transcribed spacer (ITS) region was amplified using the ITS1 (5′-CTTGGTCATTTAGAGGAAGTAA-3′) and ITS2 (5′-GCTGCGTTTCTTCATCGATGC-3′) primer set [23]. Metabarcoding of the 16S rRNA gene and ITS region was conducted according to the Illumina protocol library preparation manual. The PCR conditions used were as follows: initial denaturation at 98 °C for 1 min, 30 cycles at 98 °C for 10 s, annealing at 50 °C for 30 s, elongation at 72 °C for 1 min, and a final extension at 72 °C for 5 min. Sequencing libraries were generated using a NEBNext Ultra DNA Library Prep Kit for Illumina (New England BioLabs, Ipswich, MA, USA). Library quality was evaluated using the Qubit 2.0 Fluorometer (Thermo Fisher Scientific, Waltham, MA, USA) and Agilent Bioanalyzer 2100 system. Sequencing was performed on an Illumina MiSeq platform, generating 300 bp, paired-end reads. Sequencing reads were first filtered using the QIIME (Quantitative Insights Into Microbial Ecology) software using the default settings. The operational taxonomic units (OTUs) were classified using a 97% nucleotide sequence similarity cutoff. The Majorbio Cloud platform was utilized for bioinformatics.

Microbial alpha diversity was assessed according to the Chao1 species richness index and Shannon diversity index. Differences in microbial community composition across the

soil salinity gradient were evaluated using non-metric multidimensional scaling (NMDS). Taxonomy was assigned using the Ribosomal Database Project classifier. Microbial ecologically relevant functions were analyzed using FAPROTAX (bacteria) and FunGuild (fungi).

### 2.3. Characterization of Root Exudates

Root-associated soil samples were extracted with ethyl-acetate, and the extracted solution was concentrated and dried by lyophilization [24]. Briefly, 500 g of each airdried soil sample was placed into a glass bottle containing 1.5 l ethyl-acetate and shaken at 200 r/min at 20 °C. Shaken samples were transferred to 50 mL centrifuge tubes and centrifuged at 5000 r/min for 15 min. The supernatant was collected and concentrated to 1–1.5 mL using a rotary vacuum evaporator and then further concentrated and dried using a lyophilizer (LGJ-50F, Songyuan Huaxing Technology Company, Beijing, China). The resultant powder was dissolved in anhydrous ethyl-acetate and transferred to an ampule for storage; each salt stress treatment was repeated 3 times.

The chemical composition of each dried sample was determined by gas chromatography—mass spectrometry (GC-MS). GC was carried out according to the following conditions: injection temperature of 250 °C and gas flow rate of 1.0 mL/min. Briefly, the oven temperature was programmed at 70 °C for 2 min, and the temperature was then raised to 280 °C for 20 min with a 10 °C/min rate of increase. MS was carried out according to the following conditions: EI acceleration voltage of 70 eV, EI temperature of 230 °C, scanning speed of 0.2 s, scanning range of 35–500 amu, and solvent delay time of 3.0 min. Agilent MassHunter 5.0 was used to analyze the mass spectra and chromatograms. The linear retention indices were determined by injection of a hexane solution containing the C7–C40 series of n-alkanes under the same operating conditions. Most constituents of the soil extract were identified by comparison of their linear retention indices (RI[a]) with values in the literature [24,25]. Further identification was made by comparison of their mass spectra with those stored in the mass spectra databases (NIST ver. 14.0). The relative concentrations of each chemical component of the soil extract were obtained by peak area normalization.

### 2.4. Determination of Soil Ion Content

The concentrations of $Na^+$, $K^+$, $Ca^{2+}$, and $Mg^{2+}$ were determined by inductively coupled plasma-atomic emission spectroscopy (ICP-AEC) [26,27]. Briefly, after air-dried soil samples were sieved through a 0.15 mm mesh, 0.5 g of the sieved soil was placed into a polytetrafluoroethylene tube with 5 mL of hydrofluoric acid and 10 mL of mixed acid (1:1 nitric acid:perchloric acid) and subsequently placed in a microwave digestion system for 90 min. Digested samples were placed on a hot plate to dry at 100 °C for 3–5 min. Dried residues were transferred to a 50 mL glass tube while the original polytetrafluoroethylene tube was washed twice with deionized water. The residues were then placed back into the original polytetrafluoroethylene tubes and diluted with 10 mL of deionized water and then determined by ICP-AEC. Means of $Na^+$, $K^+$, $Ca^{2+}$, and $Mg^{2+}$ were calculated from triplicates.

The concentrations of $SO_4^{2-}$ and $Cl^-$ were determined by ion chromatography. Briefly, after air-dried soil samples were sieved through a 0.15 mm mesh, 10 g of the sieved soil was placed into a glass tube with 100 mL of double-distilled water and shaken for 2 h at 100 r/min at 25 °C, ultrasonicated for 60 min, and finally centrifuged at 5000 r/min for 15 min. The supernatant was sequentially filtered through 0.45 and 0.22 μm filter membranes. An anion separation column (Dionex IonPac TM AS18) with KOH (22 mmol/L) solution as mobile phase was used to separate $SO_4^{2-}$ and $Cl^-$ anions. The standard curve was as follows: dilute the known concentrations of $SO_4^{2-}$ and $Cl^-$ standard solutions, mix the standard solution, wash it with the prepared $Na_2CO_3$-$NaHCO_3$ to a certain volume, and measure it under the set chromatographic conditions (flow rate of 2.0 mL/min, column temperature of 30 °C, suppression current of 30 mA, and injection volume of 100 μL). The concentration of $SO_4^{2-}$ and $Cl^-$ was calculated based on the standard curve [26,27].

### 2.5. Soil Physicochemical Property Analysis Determination

Total N was determined using the micro-Kjeldahl method; total P and K were measured using the flame photometer method; the organic matter contents were determined by the potassium bichromate titrimetric method; soil pH was measured with a pH meter.

### 2.6. Statistical Analysis

Statistical differences in microbial diversity and relative abundance across the soil salinity gradient were examined by both analyses of variance (ANOVA) and Dunnett's posthoc test for multiple comparisons using SPSS ver. 16.0. Prior to conducting statistical analysis, OTU data were subjected to square root transformation, and relative abundance data were subjected to arcsine square root transformation. Additionally, the assumptions of data normality and homogeneity were checked prior to conducting statistical analysis. The effects of root exudates and soil ionic concentration on the soil microbial community were examined by multivariate redundancy analysis (RDA) using Canoco 5. Statistically significant differences and highly statistically significant differences were confirmed at the 5% ($p$-value < 0.05) and 1% ($p$-value < 0.01) levels, respectively.

## 3. Results

### 3.1. High Salt Stress Reduced the Diversity of Bacterial and Fungal Communities

Overall, the number of OTUs, the species richness (Chao1), and the species diversity (Shannon index) of soil microbes were significantly affected by salt stress (Figure 1). For bacteria, the number of OTUs ranged from 3765 ± 283 in the LS treatment to 2773 ± 274 in the HS treatment, representing a decrease of 26.3% (Figure 1A). The Shannon index suggested that the bacterial community diversity was similar in the HS and MS treatment (Figure 1C). The Chao1 index suggested that bacterial abundance was highest in the LS treatment and lowest in the HS treatment. For fungi, the number of OTUs ranged from 863 ± 83 in the LS treatment to 441 ± 74 in the HS treatment, representing a decrease of 48.9% (Figure 1D). Both community diversity (Shannon index) and abundance (Chao1) were highest in the LS treatment and lowest in the HS treatment. NMDS analysis showed that each soil salinity level was characterized by a distinct microbial community structure (Figure S1).

### 3.2. The Dominant Microbial Phyla Were Stable under Salt Stress

The dominant bacterial phyla across all soil salinity levels were Proteobacteria, Acidobacteriota, Actinobacteriota, and Chloroflexi, which together accounted for approximately 70% of the total bacterial sequences with no significant differences between treatments (Figures 2A and 3A). In fact, 38 dominant bacterial phyla were shared across all treatments, accounting for 88.37% of the total community (Figure 3A). Additionally, the community abundance of the seven core phyla was similar between treatments, and the distribution of dominant phyla in different treatments was concentrated (Figure 3C). However, significant differences between treatments were seen at the species level. As salinity increased, the relative abundance of *Geminicoccaceae*, *Rokubacteriaces*, *Comanonadaceae*, UTCFX1, *Gemmatimonadaceae*, and TK-10 increased. Conversely, as salinity decreased, *Xanchobacteraceae*, JG30-KF-AS9, *Pedomicrobium Bauidia*, *Acidibacter*, and *Hyphomicrobium* increased in relative abundance.

The dominant fungal phyla across all soil salinity levels were Ascomycota and Basidiomycota with no significant differences between treatments (Figures 2B and 3D). A total of 11 dominant fungal phyla were shared across all treatments, accounting for 84.61% of the total community (Figure 3B). Again, significant differences were seen at the species level. As salinity increased, *Vagum*, *Nemalophilum*, *Funneliformis*, and *Candida* increased in abundance. As salinity decreased, *Chytridiomycota*, *Simmonsii*, and *Trichoderma_asperellum* increased in abundance (Figure 4).

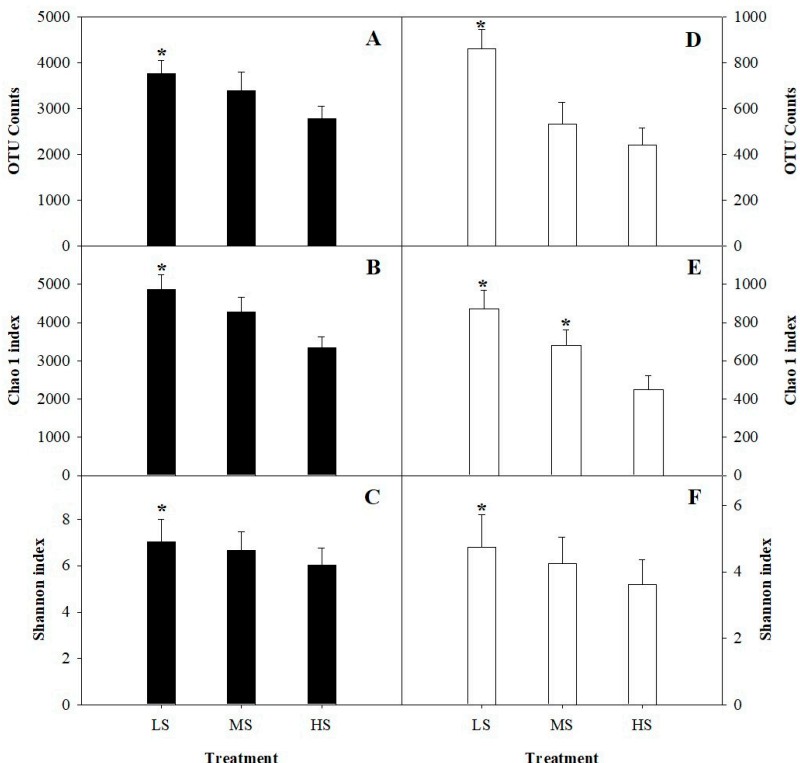

**Figure 1.** Estimated number of observed OTU counts, Shannon index, and Chao 1 index of rhizosphere microbiome. Values are means ± s.e. (*n* = 3). The star (*) indicates significant difference between treatments at 0.05 level. LS: low-salinity soil; MS: medium-salinity soil; HS: high-salinity soil. (**A–C**): the bacterial community index; (**D–F**): the fungi community index; (**A,D**): OTU counts; (**B,E**): Chao 1 index; (**C,F**): Shannon index.

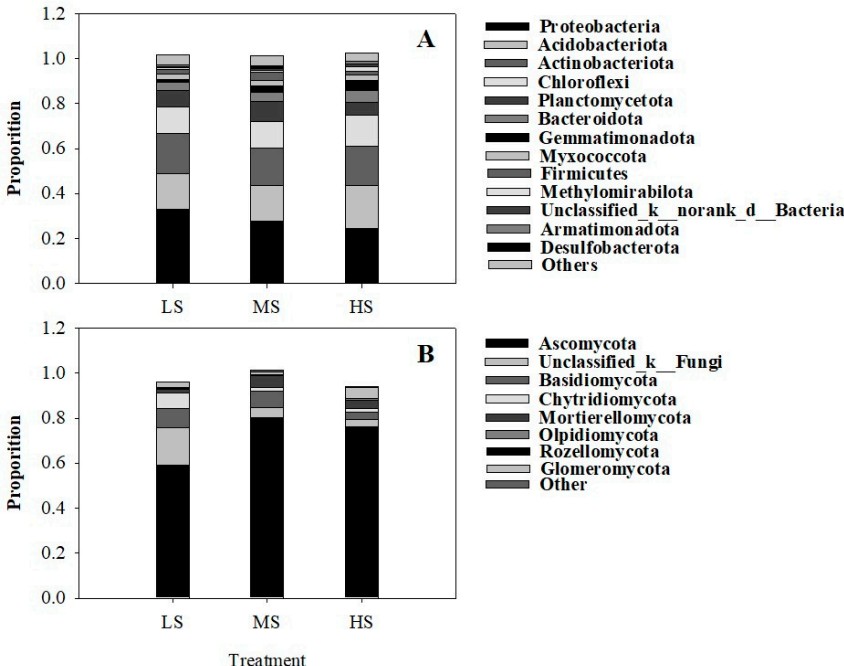

**Figure 2.** The relative abundance of phyla of rhizosphere microbiome (*n* = 3, % of total sequence). (**A**): bacterial community; (**B**): fungi community. LS: low-salinity soil; MS: medium-salinity soil; HS: high-salinity soil.

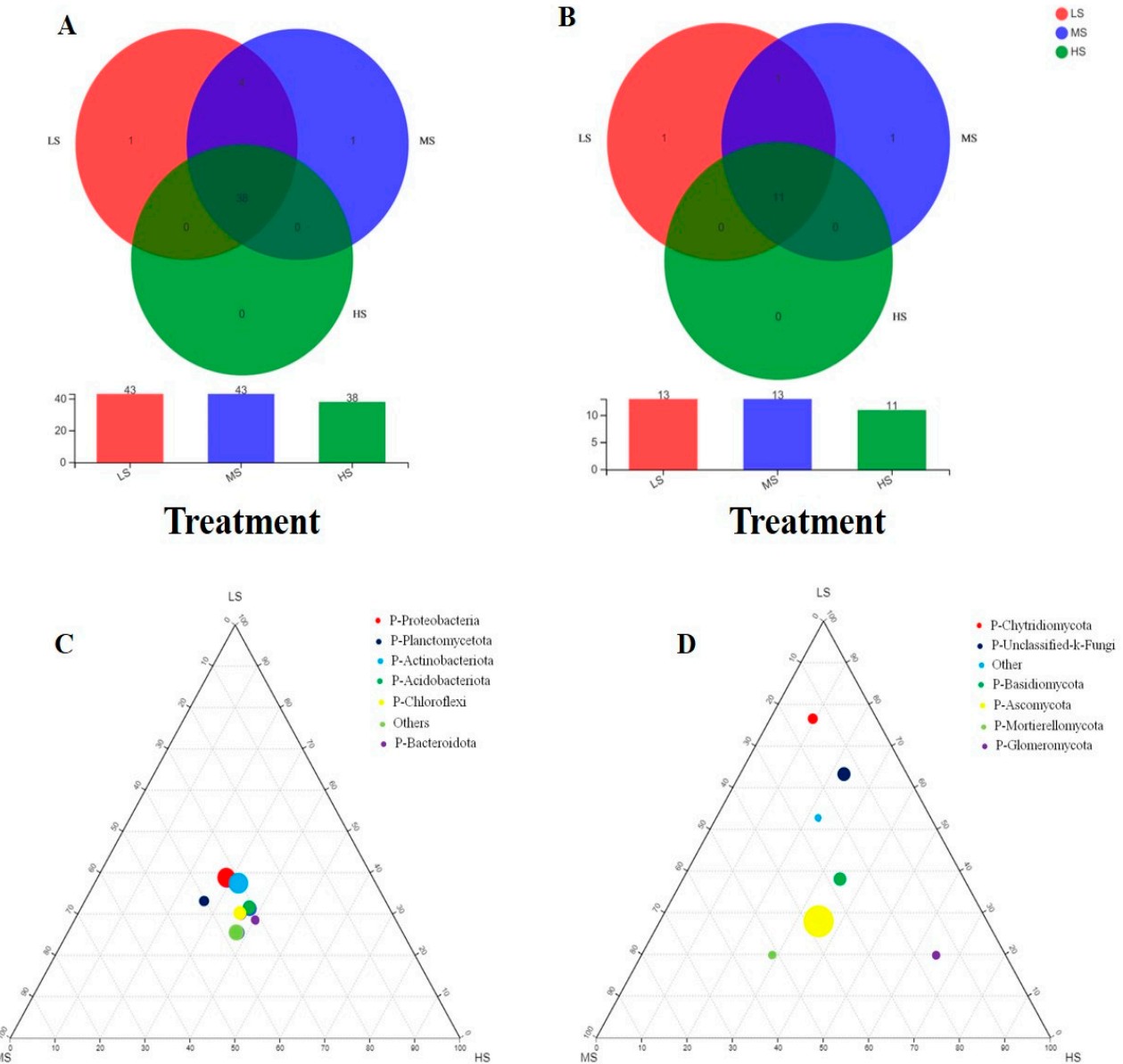

**Figure 3.** The microbial dominant phylum distribution among different saline–alkali stress treatments. (*n* = 3). (**A**): number of bacterial communities; (**B**): number of fungi communities, (**C**): core bacterial phylum; (**D**): core fungi phylum. LS: low-salinity soil; MS: medium-salinity soil; HS: high- salinity soil.

### 3.3. Most Microbial Functional Groups Were Stable under Salt Stress

Across soils, bacterial ecological functions included aromatic compound degradation, dark oxidation of sulfur compounds, manganese oxidation, nitrogen fixation, denitrification, cellulolysis, and ureolysis (Table 2). There were no significant differences between treatments for all functional groups with the exception of manganese oxidation, cellulolysis, nitrogen fixation, and ureolysis. The functional group of Manganese oxidation was most prevalent in the LS treatment, and the functional group of cellulolysis was most prevalent in the MS treatment.

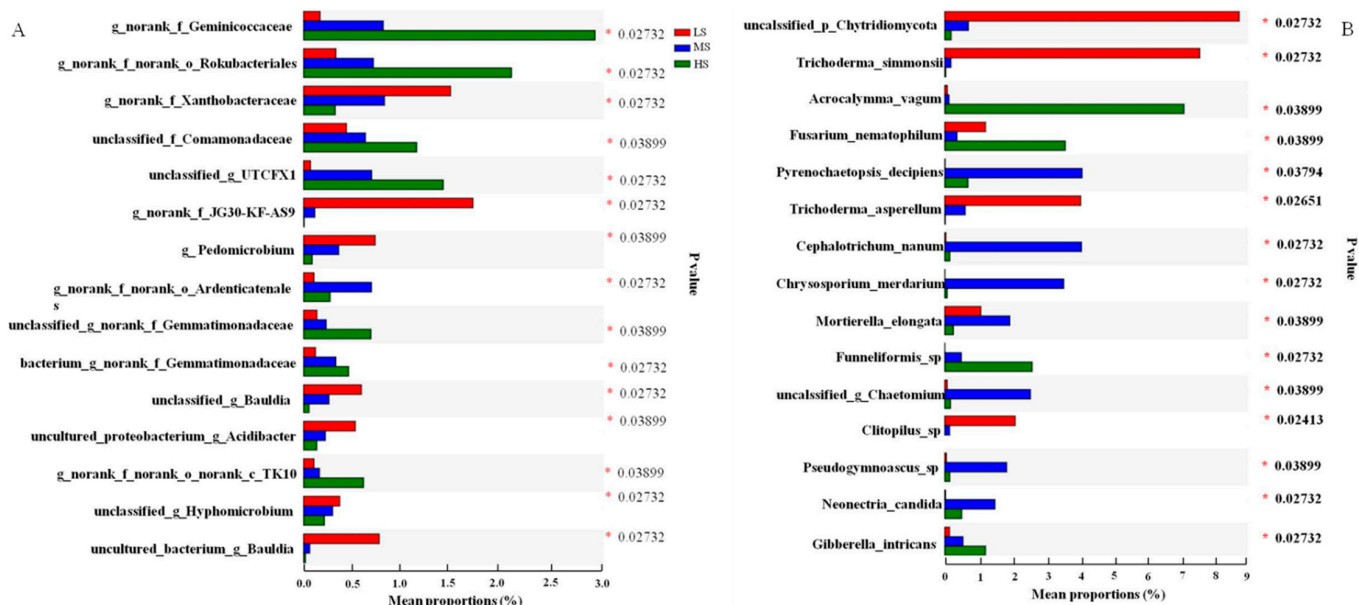

**Figure 4.** The different rhizosphere microbial species under saline–alkali stress treatment. (*n* = 3). The star (*) indicates significant difference between treatments at 0.05 level. (**A**): bacterial community; (**B**): fungi community. LS: low-salinity soil; MS: medium-salinity soil; HS: high-salinity soil.

**Table 2.** The potential function of soil bacteria (*n* = 3).

| Functional Groups | LS | MS | HS |
|---|---|---|---|
| Ureolysis | 299 ± 86 | 320 ± 34 | 490 ± 66 ** |
| Aromatic compound Degradation | 295 ± 71 | 368 ± 88 | 257 ± 63 |
| Nitrate reduction | 321 ± 80 | 228 ± 62 | 300 ± 53 |
| Nitrogen fixation | 116 ± 27 | 355 ± 58 * | 118 ± 43 |
| Cellulolysis | 103 ± 24 | 308 ± 75 * | 71 ± 17 |
| Chitinolysis | 167 ± 35 | 58 ± 12 | 229 ± 52 |
| Nitrification | 28 ± 9 | 5 ± 1 | 125 ± 39 ** |
| Aromatic hydrocarbon degradation | 7 ± 1 | 16 ± 3 | 4 ± 1 |
| Denitrification | 60 ± 15 | 95 ± 26 | 90 ± 31 |
| Hydrocarbon degradation | 42 ± 9 | 19 ± 7 | 5 ± 2 |
| Dark iron oxidation | 5 ± 2 | 5 ± 1 | 13 ± 4 |
| Dark oxidation of sulfur compounds | 21 ± 8 | 13 ± 5 | 13 ± 4 |
| Manganese oxidation | 61 ± 15 * | 22 ± 9 | 26 ± 6 |
| Nitrate denitrification | 52 ± 21 | 88 ± 34 | 25 ± 7 |
| Xylanolysis | 25 ± 5 | 17 ± 4 | 4 ± 1 |
| Photoheterotrophy | 226 ± 72 | 249 ± 63 | 68 ± 27 |
| Phototrophy | 390 ± 90 | 409 ± 102 | 149 ± 58 |

LS: low-salinity soil; MS: medium-salinity soil; HS: high-salinity soil. The star (* and **) indicates significant difference between treatments at 0.05 and 0.01 level.

Across soils, fungal ecological functions included arbuscular mycorrhizal, plant pathogen, undefined saprotroph, and endophyte (Table 3). In particular, the arbuscular mycorrhizal fungal (AMF) community was highly significantly affected by salt stress (Table 3, *p* < 0.01). As salinity increased, *Funneliformis*, *Diversispora*, *Diversispora ipigaea*, *unclassified Glomeraceae*, and *unclassified Rhizophagus* increased, and as salinity decreased, *Paraglomus* and *Glomeromycota* increased. Strikingly, the number of OTUs of *Funneliformis* increased from 1 in the LS treatment to 189 ± 40 in the MS treatment and further to 1577 ± 146 in the HS treatment. Changes in *Neonectria*, *Neocosmospora*, and *Chaetomium* were similar to that in *Funneliformis*.

**Table 3.** The potential function of soil fungi (*n* = 3).

| Guild | Phylum | Species | LS | MS | HS |
|---|---|---|---|---|---|
| Arbuscular Mycorrhizal | *Glomeromycota* | *Funneliformis* | 1 | 189 ± 40 * | 1577 ± 146 ** |
| | | *Unclassified* | 8 ± 2 | 175 ± 35 * | 972 ± 98 ** |
| | | *Paraglomus* | 406 ± 56 ** | 0 | 0 |
| | | *Diversispora* | 5 ± 1 | 0 | 45 ± 13 ** |
| | | *Diversispora ipigaea* | 5 ± 3 | 0 | 65 ± 19 ** |
| | | *Unclassified diversisiporaceae* | 1 | 0 | 55 ± 15 ** |
| | | *Unclassified glomeraceae* | 0 | 0 | 50 ± 18 ** |
| | | *Unclassified rhizophagus* | 17 ± 1 | 40 ± 11 * | 42 ± 13 * |
| | | *Glomeromycota* | 47 ± 9 ** | 0 | 0 |
| Plant Pathogen | *Ascomycota* | *Neonectria* | 15 ± 3 | 535 ± 78 * | 359 ± 64 |
| | | *Chrysanthemicola* | 326 ± 57 | 225 ± 31 | 164 ± 27 |
| | | *Ceratorhiza* | 997 ± 142 ** | 0 | 0 |
| | | *Clonostachys* | 16 ± 3 | 10 ± 2 | 6 ± 2 |
| | | *Didymella* | 47 ± 11 | 32 ± 9 | 378 ± 63 ** |
| | | *Fusarium* | 1153 ± 187 * | 266 ± 92 | 2938 ± 265 ** |
| | | *Unclassified* | 0 | 665 ± 59 ** | 50 ± 17 * |
| Undefined Saprotroph | *Ascomycota* | *Neocosmospora* | 283 ± 85 | 543 ± 152 | 2693 ± 408 ** |
| | | *Unclassified* | 1357 ± 64 ** | 93 ± 19 * | 22 ± 7 |
| | | *Trichoderma* | 1328 ± 199 ** | 423 ± 22 ** | 23 ± 8 |
| | | *Cephalotrichum* | 34 ± 7 | 2203 ± 275 * | 120 ± 14 |
| | *Basidiomycota* | *Clitopilus* | 1838 ± 150 ** | 33 ± 2 ** | 0 |
| | | *Unclassified* | 361 ± 33 | 195 ± 23 | 5 ± 1 |
| Endophyte | *Ascomycota* | *Chaetomium* | 48 ± 8 | 2481 ± 553 * | 104 ± 46 |
| | | *Unclassified* | 905 ± 79 * | 589 ± 47 | 390 ± 15 |
| | *Mortierellomycota* | *Mortierella* | 504 ± 44 | 357 ± 85 | 180 ± 31 |
| Ericoid Mycorrhizal | *Ascomycota* | *Oidiodendron* | 151 ± 34 ** | 1 | 1 |

LS: low-salinity soil; MS: medium-salinity soil; HS: high-salinity soil. The star (* and **) indicates significant difference between treatments at 0.05 and 0.01 level.

### 3.4. Salt Stress Altered the Composition of Root Exudates

Across all salinity levels, the most common constituents of the chamomile root exudate were 4-(2′, 4′, 4′-trimethyl-yciclo [4.1.0] hept-2′-en-3′-yl)-3-buten-2-one, (−)-isocomene, cubenol, and (*E*)-β-famesene (Table 3). The rarest constituents were methyl 4-methylbenzoate, dodecanoic acid, methyl ester, and methyl isomyristate. Salt stress was found to significantly decrease the content of (−)-isocomene in root exudates. In the LS treatment, the (−)-isocomene content was 41.43 ± 9.61%, while it was 19.14 ± 1.80% and 25.28 ± 4.61% in the MS and HS treatments, respectively. Conversely, salt stress was found to significantly increase the content of (*E*)-β-famesene. Interestingly, the cubenol content was highest in the MS treatment with a content of 1.19 ± 0.04%, 16.94 ± 3.49%, and 5.21 ± 1.47% in the LS, MS, and HS treatments, respectively (Table 4).

### 3.5. $SO_4^{2-}$, $Ca^{2+}$, and $Na^+$ Were the Dominant Salt Ions

The concentration of $SO_4^{2-}$, $Ca^{2+}$, and $Na^+$ increased as overall salinity increased, accounting for 88.10%, 92.17%, and 93.33% of the total ionic content of LS, MS, and HS soils, respectively (Table 4). Across soils, the concentration of $Cl^-$ and $K^+$ were low, accounting for only 2.26%, 1.96%, and 1.43% of the total ionic content of LS, MS, and HS soils, respectively. In both cases, there was no significant difference between treatments (Table 4).

**Table 4.** Chemical composition of root exudation in different soil saline–alkali treatments (*n* = 3).

| Peaks | Name | RI | | RT | | | Peak Area (%) | | |
|---|---|---|---|---|---|---|---|---|---|
| | | RI [a] | RI [b] | LS | MS | HS | LS | MS | HS |
| 1 | 4-Oxo-4-(para-tolyl)-butyric acid | | 993.6 | / | 8.12 | 8.12 | / | 1.83 ± 0.24 * | 0.29 ± 0.01 |
| 2 | Methyl 4-methylbenzoate | | 1095.3 | 13.83 | 13.84 | 13.83 | 1.49 ± 0.35 | 1.5 ± 0.5 | 0.99 ± 0.11 |
| 3 | 1,2,4,8-Tetramethylbicyclo [6.3.0] undeca-2,4-diene | | 1347.7 | 18.38 | 18.38 | 18.38 | 4.27 ± 1.82 | 1.69 ± 0.47 | 2.94 ± 0.57 |
| 4 | 4- (2′, 4′, 4′-trimethyl-yciclo [4.1.0] hept-2′-en-3′-yl)-3-buten-2-one | | 1384.6 | 19.26 | 19.27 | 19.27 | 16.58 ± 2.78 | 17.53 ± 3.19 | 14.22 ± 2.53 |
| 5 | (−)-Isocomene | 1387 | 1391.3 | 19.43 | 19.43 | 19.43 | 41.43 ± 9.61 * | 19.14 ± 1.80 | 25.28 ± 4.61 |
| 6 | (±) -β-Isocomene | 1407 | 1411.4 | / | 19.89 | 19.89 | 6.9 ± 0.78 | / | 5.46 ± 0.35 |
| 7 | β-Caryophyllene | 1419 | 1424.5 | 20.17 | 20.17 | 20.18 | 1.83 ± 0.70 | 5.44 ± 0.87 | 4.85 ± 0.48 * |
| 8 | (E) -β-Famesene | 1444 | 1458.4 | 20.92 | 20.92 | 20.92 | 8.36 ± 1.06 | 16.82 ± 6.04 * | 15.37 ± 3.86 |
| 9 | γ-Muurolene | 1477 | 1464.8 | 21.07 | 21.07 | 21.07 | 2.79 ± 0.99 | 2.11 ± 0.29 | 3.03 ± 0.37 |
| 10 | Germacrene D | 1480 | 1480 | 21.40 | 21.40 | 21.40 | 9.36 ± 1.25 | 5.12 ± 1.17 | 11.8 ± 3.21 |
| 11 | δ-Selinene | 1492 | 1484 | / | 21.49 | 21.49 | 0.94 ± 0.08 | / | 1.26 ± 0.07 |
| 12 | Valencen | 1496 | 1488.3 | / | 21.58 | 21.59 | 0.74 ± 0.03 | / | 1.19 ± 0.07 |
| 13 | β-Bisabolene | 1509 | 1510.7 | 22.07 | 22.08 | 22.08 | 0.45 ± 0.02 | 0.41 ± 0.01 | 0.33 ± 0.01 |
| 14 | Dodecanoic acid, methyl ester | | 1522.9 | 22.35 | 22.35 | 22.34 | 0.18 ± 0.01 | 3.8 ± 0.03 * | 0.3 ± 0.01 |
| 15 | Trans-Calamenene | 1529 | 1527.2 | 22.43 | 22.44 | 22.43 | 0.32 ± 0.01 | 1.32 ± 0.07 | 0.45 ± 0.06 |
| 16 | Cubenol | 1642 | 1639.4 | 24.90 | 24.90 | 24.90 | 1.19 ± 0.04 | 16.94 ± 3.49 * | 5.21 ± 1.47 |
| 17 | Methyl-12-methyltridecanoate | | 1687.9 | 25.99 | 25.99 | 25.99 | 0.09 ± 0.01 | 3.58 ± 0.98 * | 0.33 ± 0.01 |

LS: low-salinity soil; MS: medium-salinity soil; HS: high-salinity soil; RI: indicates retention index; [a]: Literature [23,24]; [b]: HP-5MS column. The star (*) indicates significant difference between treatments at 0.05 level. The mark "/" means that the component was not detected in this sample.

### 3.6. Bacteria Were Sensitive to Salinity, while Fungi Were Sensitive to Root Exudates

RDA indicated that the bacterial community tended to be more sensitive to changes in salinity, while the fungal community tended to be more sensitive to changes in root exudates. Specifically, the main explanatory environmental factors were found to be $SO_4^{2-}$, $Ca^{2+}$, $Na^+$, SE (δ-selinene), and ME (methyl 4-methylbenzoate), together explaining >95% of the variance (Table S1). Several environmental factors and bacterial groups were clustered in the fourth quadrant, including $SO_4^{2-}$, $Ca^{2+}$, $Na^+$, $Mg^{2+}$, SE, VA (valencen), ISO ((±) -β-isocomene), CA(β-caryophyllene), Acidobacteriota, Gemmatimonadota, Methylomirabilota, and Firmicutes (Figure 5). Other environmental factors and microbial groups were clustered in the second quadrant, including BIS (β-bisabolene), IS ((−)-isocomene), the bacterial groups Proteobacteria and Actinobacteriota, and the fungal groups Chytridiomycota, Basidiomycota, and Olpidiomycota.

Species-level RDA showed that several environmental factors and microbial groups were clustered in the first and second quadrants, including $SO_4^{2-}$, $Ca^{2+}$, $Na^+$, ISO, CA, FA((*E*)-β-Famesene), SE, the bacterial groups *Geminicoccaceae*, *Rokubacteriales*, *Comamonadaceae*, UTCFX1, *U-Gemmadimonadaceae*, and *B-Gemmatimonadacea*, and the fungal groups *Vagum*, *Funneliformis*, *Intricans*, and *Nematophilum*. Several other environmental factors and microbial groups clustered in the fourth quadrant, including TY(4-(2′, 4′, 4′-trimethyl-yciclo [4.1.0] hept-2′-en-3′-yl)-3-buten-2-one), ME, BIS, the bacterial groups *Xanthobacteraceae*, JG30-KT-AS9, *Pedomicrobium*, *U-Bauldia*, *Acidibacter*, *Hyphomicrobium*, and *B-Bauldia*, and the fungal groups *Chytridiomycota*, *Simmonsii*, *Asperellum*, and *Clitopilus*.

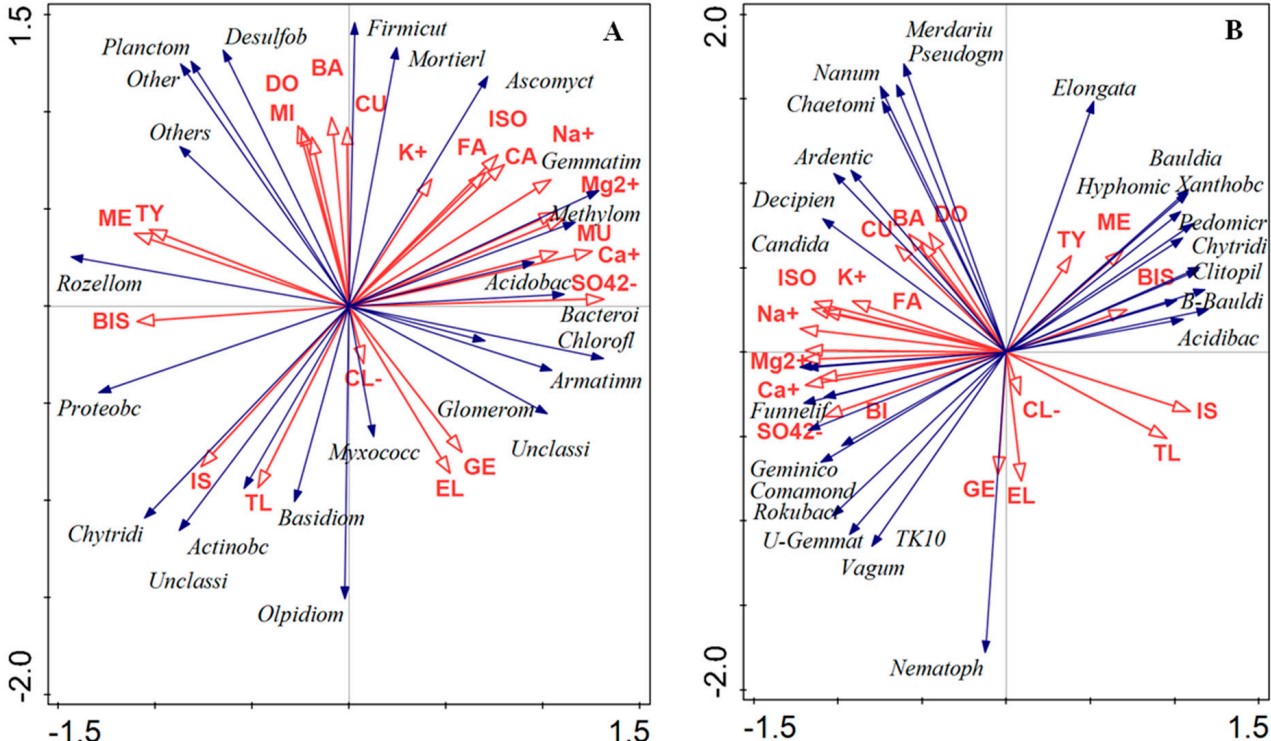

**Figure 5.** Redundancy analysis of soil microbial and environmental factors. (**A**): based on the phyla; (**B**): based on the species. SE: δ-Selinene, ME: Methyl 4-methylbenzoate, TY: 4-(2′, 4′, 4′-trimethyl-yciclo [4.1.0] hept-2-en-3′-yl)-3-buten-2-one, BIS: β-Bisabolene, VA: Valencen IS: (−)-Isocomene, ISO: (±) -β-Isocomene, CA: β-Caryophyllene, TL: 1,2,4,8-Tetramethy lbicyclo [6.3.0] undeca-2,4-diene, MU: γ-Muurolene, FA: (*E*)-β-Famesene, BA: 4-Oxo-4- (para-tolyl)-butyric acid, DO: Dodecanoic acid, methyl ester, GE: Germacrene D, MI: Methyl-12-methyltridecanoate, CU: Cubenol, TC: trans-Calamenene.

Comprehensive analysis of the relationship between soil salt ions, root exudate compounds, and microbial communities indicated that fungi and root exudate compounds tended to cluster together in the first, second, and third quadrants. Even in the fourth quadrant, the fungal groups Ascomycota and Moriterllomycota were most strongly associated with root exudates (Figure 5A). The relationship between fungi and root exudates was stable at the species level as well with the fungal species *Chytridiomycota*, *Simmonsii*, *Asperellum*, *Clitopilus*, *Decipiens*, *Cephalotrichum-nanum*, *Merdarium*, *Chaetomium*, and *Candida* clustering primarily with root exudates in the first and fourth quadrants. Salt ions were primarily clustered in the first and second quadrants and displayed a strong relationship with bacterial groups rather than fungal groups.

## 4. Discussion

### 4.1. Salt Stress Shapes Diversity and Community Structure of Rhizospheric Microbiota in Chamomile

Soil microbes play an important role in regulating the plant rhizosphere environment and enhancing plant salt stress tolerance by providing a buffer zone for plants against stress, producing various plant growth-promoting hormones, promoting nutrient cycling and organic matter decomposition [5]. So, the stable bacterial and fungal community structure of the rhizosphere is particularly important [4]. In this work, we found that salt stress significantly altered both the diversity and community structure of soil microbes, although the dominant rhizosphere phyla were little unaffected (Figures 1 and 2). Across

salinity levels, we found 38 dominant bacterial phyla and 11 dominant fungal phyla, accounting for 88.37% and 84.61% of the community, respectively (Figure 3). These results indicate that the core microbial community of the chamomile rhizosphere is relatively stable under salt stress.

In some cases, the ecologically relevant microbial functional groups were also significantly altered by salt stress. For bacteria, functional groups of chitinolysis, hydrocarbon degradation, denitrification, and aromatic compound degradation were found to be unaffected by salt stress, while functional groups of nitrogen fixation, nitrification, and ureolysis tended to increase under MS and HS treatments (Tables 2 and 3). Previous studies have found that salinity-induced osmotic stress was toxic to soil microbes, indirectly affecting soil nitrogen transformation and absorption [28]. In wheat, inoculation with *Bacillus aquimaris* increased the N content of leaves and promoted plant growth under salt stress [29]. The abundance of AMF also increased significantly in the HS treatment, including *Funneliformis*, *unclassified Diversisiporaceae*, *Diversispora ipigaea*, and *Diversispora* (Table 1). Sziderics et al. showed that in tomato, AMF increased the content of osmoregulatory sugars and amino acids in plant cells, increasing water absorption, nutrient uptake, and growth [2,7].

Overall, the bacterial and fungal groups with increased community abundance accounted for 53.33% and 73.33% of the total rhizosphere microbial species, respectively (Figure 4). Microbes that are themselves tolerant of salt stress may provide increased salt tolerance to their host plants. Thus, we speculated that the tolerance of chamomile to salt stress may be related to the stability of the core microbial community and the altered functions associated with salt-tolerant rhizosphere microbes.

### 4.2. The Relationship between Environment Factors and the Rhizosphere Microbial Community

Environmental conditions, such as root exudates, soil ions, pH, etc., affected the diversity and community structure of the rhizosphere microbes. Root exudates had multiple effects, such as antimicrobial, attractant, or promoting effects, and soil ions had osmotic and toxicity effects. In our experiment, although salt stress had little effect on the dominant phyla, individual species were significantly affected (Figure 4). For example, the abundances of the bacterial species *Xanthobacteraceae* and *Pedomicrobium* and the fungal species *Chytridiomycota*, *Simmonsii*, and *Asperellum* were lowest in the HS treatment ($p < 0.05$) (Figure 4). These results may be due to the antimicrobial activity of certain root exudates. Previous studies indicate that several compounds found in root exudates, including SE, ME, BIS, β-caryophyllene, cubenol, and (±)-β-isocomene, have antimicrobial properties [18,19]. Additionally, rosmarinic acid from the root exudates of sweet basil has antibacterial activity against *Pseudomonas aeruginosa* [30]. RDA indicated that the root exudate compounds TY, ME, and BIS were closely related to the abundances of these species (Table 4, Figure 5B). However, most microbial species with decreased abundances under salt stress were more sensitive to the presence of salt ions, particularly $SO_4^{2-}$ and $Ca^{2+}$ (Table 5). High salinity increased the osmolarity of the aqueous solution surrounding the microbial cell, producing toxicity and limiting microbial activity [10,11]. It appears that both the antibacterial compounds present in the chamomile root exudate and the high concentration of certain salt ions effectively reduce the abundance of certain sensitive rhizosphere microbes.

**Table 5.** The saline–alkali ion content in different soil saline–alkali treatments ($n = 3$).

| Treatment | $SO_4^{2-}$ (cmol/kg) | $Cl^-$ (cmol/kg) | $Na^+$ (mg/kg) | $K^+$ (mg/kg) | $Ca^{2+}$ (mg/kg) | $Mg^{2+}$ (mg/kg) |
|---|---|---|---|---|---|---|
| LS | 0.11 ± 0.03 | 0.04 ± 0.01 | 0.76 ± 0.13 | 0.04 ± 0.01 | 2.24 ± 0.09 | 0.34 ± 0.01 |
| MS | 0.38 ± 0.09 | 0.03 ± 0.01 | 3.79 ± 0.90 * | 0.14 ± 0.05 | 3.84 ± 0.35 | 0.51 ± 0.03 |
| HS | 0.89 ± 0.21 * | 0.04 ± 0.01 | 3.87 ± 0.14 * | 0.11 ± 0.01 | 5.04 ± 1.15 * | 0.55 ± 0.02 |

LS: low-salinity soil; MS: medium-salinity soil; HS: high-salinity soil. The star (*) indicates significant difference between treatments at 0.05 level.

Interestingly, the abundances of other bacterial species, including *Geminicoccaceae, Rokubacteriaces, Comanonadaceae*, and UTCFX1, and fungal species, including *Vagum, Nemalophilum, Funneliformis*, and *Candida*, increased under salt stress (Figure 4). This may be related to the inherent tolerance of certain microbes to salt toxicity [31]. Previous work has found that high soil salinity results in an increase in the relative abundance of *Gemmatimonadetes* and *Bacteroidetes* [4,13]. Additionally, roots may also exude stimulatory substances that can promote the growth of rhizosphere microbes, including sugars, amino acids, and aromatic compounds, such as shikimate, quinate, protocatechuate, vanillate, acetosyringone, gallate, catechol, and luteolin [32]. The carbon present in many of these compounds can also be used as a nutrient by many rhizospheric microbes [33–35]. It appears that the relationship between soil ion content, plant root exudates, and the rhizosphere microbial community is a complex one.

**5. Conclusions**

Salt stress led to changes in the rhizosphere microbial community structure without altering the identities or abundances of the dominant phyla. The salt stress adaptability of chamomile may be related to the relative stability of the core rhizosphere phyla, as well as the increased presence of salt-tolerant microbes and their associated ecological functions under salt stress. Bacterial abundance was primarily affected by soil ionic content, particularly $SO_4^{2-}$, $Ca^{2+}$, $Na^+$, and $Mg^{2+}$. Fungal abundance was primarily affected by the presence of certain root exudate compounds, including methyl 4-methylbenzoate, δ-selinene, and (-)-isocomene.

**Supplementary Materials:** The following supporting information can be downloaded at: https://www.mdpi.com/article/10.3390/agronomy13061444/s1, Figure S1: Non-metric Multidimensional Scaling (NMDS) of microbial communities based on OTUs for all soil samples; Table S1: Redundancy analysis (RDA).

**Author Contributions:** Conceptualization, L.S.; Data curation, F.X., H.H. and Y.Q.; Formal analysis, H.H.; Funding acquisition, H.L., Z.S. and L.S.; Investigation, F.X. and H.B.; Methodology: F.X. and Y.Q.; Project administration: L.S. and Z.S.; Writing—original draft, H.H and F.X.; Writing—review and editing, L.S., H.L., F.X. and H.H. contributed equally to this work. All authors have read and agreed to the published version of the manuscript.

**Funding:** This study was funded by supported by grants from the National Key R&D Program of China (2019YFD1002701), the Key Research Program of the Chinese Academy of Sciences (Grant NO. KFZD-SW-113), and Key R&D Projects in Hebei Province (Grant NO. 22327103D). The APC was funded by the Key Research Program of the Chinese Academy of Sciences (Grant NO. KFZD-SW-113).

**Institutional Review Board Statement:** Not applicable.

**Data Availability Statement:** The datasets used during the current study are available from the corresponding author on reasonable request.

**Acknowledgments:** The authors duly acknowledge the support received from the Agricultural High-tech Industrial Demonstration area of the Yellow River Delta of Shandong Province.

**Conflicts of Interest:** The authors declare no conflict of interest.

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
