# Peer review of "Effect of Salt Stress on Microbiome Structure and Diversity in Chamomile (Matricaria chamomilla L.) Rhizosphere Soil"

_agronomy, doi:10.3390/agronomy13061444_

Round 1

Reviewer 1 Report

Dear Authors,

Research on microbiome structure and diversity of rhizosphere soil under salt stress is a matter of many studies. Overall the study is well performed, and quite well understood. The methods chosen by the authors are appropriate and well-described. Data have been provided and they are statistically sound.Conclusions are generally supported by data. However, there are a few issues with the methodology, results and their presentation that I would suggest to be improved.

- I have doubt about centrifugation of the samples (liines 106-109) for 15 min at 5,000 r/min (this speed results with setting down soil particles and cells of microbes) and it was a material used for the extraction of DNA but it is not clear for me what happened to the supernatant and what for it was stored at -80 C.

- In case of the primer nucleotide sequences, I would suggest to ad information about 5' and 3' ends. I would also suggest to change "validate" to "verify" the integrity of the DNA (line 114)

- It should be confirmed that the genera "Asperellum" occur - I have found species "Trichoderma asperellum" and no information about "Asperellum spp."

-  It is not stated what "/" (table 3) means

- Figures scales should be adjusted to fit the data (e.g.  Figures 4 and 5)

- In Figure 4, the presentation of the microbial classification could be improved (to make it more clear for the reader)

- typos and punctuation should be checked as well as the size of the font (like in line 92-93) in the entire text

Reviewer 2 Report

The manuscript (MS) evaluates the influence of salinity on the community of microorganisms that colonize the rhizosphere of chamomile plants (xxx xxx) growing at different salt concentrations. The influence of plant exudates on the diversity of the rhizosphere microorganisms is also evaluated. The work is generally well performed, and the results are reasonably well discussed. The manuscript should be revised for clarity of language and correction of misspelled words.

However, the weak point of the manuscript lies in the importance of the study and in several points associated with it:  1- Is chamomile grown in areas with salinity problems? This point was not made clear in the introduction. It is stated in the MS that this species has a high tolerance to salinity, when above 100 mM several studies point to a decrease in growth. 2- Why is sediment from the Yellow River delta chosen for the experiment when chamomile is a species that grows best in well-drained soils? 3- The communities of microorganisms present in the sediments of the Yellow River delta will certainly be different from the communities in soils where chamomile is normally cultivated. Therefore, there will be reservations in considering the results obtained relevant and it will be difficult to extrapolate the results obtained to soils where chamomile is normally cultivated.

Additionally some specific comments were made that need to be explained or changed:

 Comment 1: The osmotic effect is not the only effect that salinity causes on micro-organisms and plants, otherwise salt stress and drought would induce identical effects. The toxicity originated directly from ions should also be addressed. 

Comment 2: Proline is not a protein.

 Comment 3: Comment 3: The grammar of this sentence needs to be corrected.

 Comment 4: Please explain what means "medicines" in the plural.

 Comment 5: This sentence has to be corrected. Chamomile does not have high tolerance to salt stress. According to reference 18 it has a high degree of tolerance to soil alkalinity and accumulates fairly large quantities of sodium. That is completely different from stating that Chamomile has high tolerance to salinity stress. In reference 19, I did not find any mention of the tolerance of this plant species to salinity. Furthermore, several studies point to chamomile as a species moderately tolerant to salinity and whose growth is affected by concentrations above 100 mM NaCl.

 Comment 6: I did not understand why the authors chose to use sediments from the Yellow River delta, since chamomile is neither a halophyte plant nor a plant that occupies waterlogged soils. On the contrary, it is a plant that grows best in well-drained soils. Authors must justify this choice.

 Comment 7: The volume of water and frequency should be described.

Comment 8: Why filter after centrifugation and then centrifuge again for the same time and at the same speed as the first centrifugation?

 Comment 9: Which were the reference materials used? The process should be better explained.

Comment 10: Were calibration curves used? The process should be better explained.

Comment 11: How do the authors know that the compounds were exuded by the plants and were not already in the sediments? The use of GC-MS only analyzes a part of the exudates, those with greater volatility. Many other compounds that are more abundant than volatiles and that are part of plant exudates are not identified by this process (most flavonoids, sugars, organic acids, amino acids, etc.)

Reviewer 3 Report

The manuscript entitled (Effect of salt stress on microbiome structure and diversity in chamomile (Matricaria chamomilla L.) rhizosphere soil) studied the salinity effect on the structure and diversity of Chamomile (Matricaria chamomilla L.) soil microbial community. The research has some points needs handling.

What is the novelty of the research, include it point by point?

What is the source of Chamomile seeds?

Chemical and physical soil properties need to add to the method

What is the irrigation rates and the moisture contents?

Whey the authors didn’t analyze also the endophytes microorganisms which is more related to the plant than the soil ones?

Introduction part needs to be more informative.

The discussion part is very poor and needs more information’s and also needs recent references.

The whole manuscript needs English editing and grammar corrections.

Round 2

Reviewer 2 Report

The answers to the specific comments were generally correct.

However, response to general comments that compromise the relevance of the work remain poorly explained:

1-    Is chamomile grown in areas with salinity problems? This point was not made clear in the introduction. It is stated in the MS that this species has a high tolerance to salinity, when above 100 mM several studies point to a decrease in growth.

2-    Why is sediment from the Yellow River delta chosen for the experiment when chamomile is a species that grows best in well-drained soils?

3-    The communities of microorganisms present in the sediments of the Yellow River delta will certainly be different from the communities in soils where chamomile is normally cultivated. Therefore, there will be reservations in considering the results obtained relevant and it will be difficult to extrapolate the results obtained to soils where chamomile is normally cultivated.

For the manuscript be accepted for publication these three topics have to be properly addressed.

Reviewer 3 Report

The authors did good work in enhancing the manuscript. I recommended accepting it after checking the reference style one by one to meet the journal requirements.
